# Clinical carbapenem-resistant *Enterobacterales* in a University Hospital in Dakar, Senegal: genomic insights into *Enterobacter hormaechei* ST182 strains carrying *bla*NDM-5 and *bla*OXA-48 genes

Komla Mawunyo Dossouvi,[1,2] Bissoume Sambe Ba,[3] Gora Lo,[1,4] Fábio Parra Sellera,[5,6] João Pedro Rueda Furlan,[7] Antoine Culot,[8] Guillaume Abriat,[8] Adja Bousso Gueye,[9] Awa Ba-Diallo,[1,4] Assane Dieng,[1] Fatime Poulo Ly,[9] Abdoulaye Cissé,[1] Serigne Mbaye Lo Ndiaye,[1] Alioune Tine,[1] Farba Karam,[1] Habsa Diagne-Samb,[1] Safietou Ngom-Cisse,[1] Halimatou Diop-Ndiaye,[1,4] Coumba Toure-Kane,[4] Aïssatou Gaye-Diallo,[1,4] Sika Dossim,[10] Souleymane Mboup,[1,4] Cheikh Saad Bouh Boye,[1] Abdoulaye Seck,[11] Makhtar Camara[1,4]

**ABSTRACT** Senegal has witnessed the emergence and spread of carbapenem-resistant *Enterobacterales* (CRE), which often cause deadly infections. Accordingly, this study aimed to determine the antimicrobial susceptibility and prevalence of carbapenemases, as well as to perform a whole-genome sequence analysis of clinical CRE isolates from a university hospital in Dakar, Senegal. MALDI-TOF MS and VITEK2 systems were used for bacterial identification and antimicrobial susceptibility testing (AST). Carbapenemase- and cephalosporinase-encoding genes were screened using simplex end-point polymerase chain reaction. Whole-genome sequencing (WGS) was performed using the Illumina MiSeq platform. The CRE isolates were resistant to almost all the 34 antimicrobials tested. Nevertheless, colistin and amikacin remained active, with susceptibility rates of 96% and 71%, respectively. Only the carbapenemase genes *bla*OXA-48 (53.8%; 15/28) and *bla*NDM (35.7%; 10/28) and the cephalosporinase gene *bla*CMY-1 (25%; 7/28) were identified. In this context, two extensively drug-resistant *Enterobacter hormaechei* isolates were subjected to WGS analysis. These isolates were assigned as sequence type (ST) 182 and carried several genes related to antimicrobial resistance (AMR), metal tolerance, and virulence. An IncL/M plasmid with 61,054 bp in length was identified as carrying the *bla*OXA-48 gene, whereas an IncFIB(pECLA)/IncFII(pECLA)/IncX3 mutireplicon plasmid with 217,745 bp in length was detected as harboring the *bla*NDM-5 gene and other genes related to AMR and metal tolerance. Our study presents the first landscape of clinical CRE circulating in Senegal, along with additional genomic analysis of *E. hormaechei* ST182 strains, which could be useful for mitigating the burden associated with CRE in this country.

**IMPORTANCE** The investigation of global critical priority CRE isolates has become crucial to reduce morbidity and mortality associated with AMR. This study revealed that colistin and amikacin can be considered good alternatives for treating CRE-associated infections in Dakar. In addition, the genomic approach revealed that the CRE isolates carried both a wide resistome and virulome. Moreover, the abundance of horizontal gene transfer regions in the genomes suggests the great implications of mobile genetic elements in the spread of AMR in Dakar. Furthermore, this study reported the complete sequences of chromosomes and *bla*OXA-48 and *bla*NDM-5-carrying plasmids. Our findings are of great importance because complete genome sequences are still rarely characterized in the West African region. Finally, this study highlights the importance of strengthening genomic surveillance of CRE in sub-Saharan African countries to mitigate the burden associated with these pathogens.

**Peer Reviewer** Beiwen Zheng, Zhejiang Chinese Medical University School of Basic Medical Sciences, Hangzhou, China

Address correspondence to Komla Mawunyo Dossouvi, dossouvikomlamawunyo@gmail.com.

The authors declare no conflict of interest.

**KEYWORDS** antimicrobial resistance, extensively drug-resistant bacteria, genomic surveillance, Africa

Carbapenems are last-resort antimicrobials used against multidrug-resistant (MDR) bacteria (1, 2). These antimicrobial agents penetrate the bacterial wall and bind irreversibly to the active site of penicillin-binding proteins (PBPs), leading to the inactivation of inhibitory autolytic intracellular enzymes and consequently the killing of bacteria (1). Several carbapenems are available, including imipenem, ertapenem, meropenem, doripenem, biapenem, tebipenem, tomopenem, razupenem, faropenem, and panipenem, with imipenem and meropenem the most used worldwide (3, 4). Worryingly, in the past decade, we have witnessed the emergence and spread of carbapenem-resistant *Enterobacterales* (CRE) (5, 6), which commonly exhibit resistance to several other antimicrobials, leading to extensively drug-resistant (XDR) or pandrug-resistant phenotypes. Accordingly, CRE constitutes a serious public health threat due to the high probability of treatment failure with fatal outcomes (7, 8). In resource-limited settings, the clinical impact of CRE is particularly more critical due to the restricted availability of advanced diagnostic tools, limited treatment options, and inadequate infection control infrastructure (9). These challenges increase the risk of therapeutic failure, prolonged hospital stays, and mortality, making CRE a critical concern in such regions (10).

In Gram-negative bacteria, carbapenem resistance is mediated mainly by the production of carbapenemases (enzymes that hydrolyze carbapenems), overexpression of efflux pumps, and impermeability due to outer membrane porin (OMP) mutations (11, 12). Additionally, plasmid-mediated cephalosporinase (CMY) combined with reduced outer membrane permeability can also confer resistance to carbapenems (11, 13). In *Enterobacterales*, carbapenemase production is the main mechanism underlying carbapenem resistance. These enzymes are grouped into the A, B, and D Ambler classes as follows: (i) class A mainly includes imipenem-hydrolyzing β-lactamase (IMI), Guiana extended-spectrum carbapenemase (GES), and *Klebsiella pneumoniae* carbapenemase (KPC) (14, 15); (ii) class B, also known as metallo-β-lactamases (MBL), mainly includes New Delhi metallo-β-lactamase (NDM), Imipenem-resistant *Pseudomonas* type carbapenemase (IMP), and Verona integron-encoded metallo-β-lactamase (VIM) (16, 17); and (iii) class D, also called OXA-type carbapenemases, mainly include OXA-48-like, OXA-23, OXA-51, OXA-62, and OXA-213 (18, 19). Carbapenemases are encoded by genes often located on plasmids (e.g., IncL/M, IncX3, IncFIIK2, IncF1A, and IncI2), class 1, 2, 3 integrons, and transposons (e.g., Tn*4401* and Tn*199*) (20, 21), leading to wide and rapid dissemination.

Although CRE strains have been a global concern, there remain significant epidemiological and genomic gaps regarding the spread of carbapenemase-producing bacteria in African countries, where resources for genomic surveillance have been more limited. This study aimed to determine the antimicrobial susceptibility and carbapenemase genes and conduct a whole-genome sequencing (WGS) analysis of clinical CRE isolates obtained in a Dakar university hospital, Senegal.

## RESULTS

### Bacterial isolates

Among the 28 CRE isolates, 17 (60.7%) were *Klebsiella pneumoniae*, seven (25%) *Enterobacter* spp., three (10.7%) *Escherichia coli*, and one (3.6%) *Proteus mirabilis*. Sixteen (57.1%) of them were isolated from outpatients (community-acquired [CA]), whereas 12 (42.9%) were isolated from inpatients (hospital-acquired [HA]). The patients from whom the strains were isolated were from the urology department (12/28; 42.9%), anesthesiology-reanimation (5/28; 17.9), orthopedics (3/28; 10.7%), pediatrics (2/28; 7.1%), internal medicine (2/28; 7.1%), external consultation (2/28; 7.1%), nephrology (1/28; 3.6%), and cardiology (1/28; 3.6%). Additionally, 16 (57.2%) were isolated from urine (UP-CRE), and

12 (42.8%) were obtained from pus, sputum, bronchial fluid, and blood (NoUP-CRE) (Table 1).

## Antimicrobial susceptibility

The 28 CRE isolates were resistant to almost all the 34 antimicrobials tested. This test revealed that 96.4% (27/28), 60.7% (17/28), and 46.4% (13/28) of the isolates were resistant to ertapenem, imipenem, and meropenem, respectively. None of the 28 CRE were susceptible to ertapenem, whereas 10.7% (3/28) were susceptible to imipenem and meropenem. The resistance rates to cephalosporins were very high, including 89.3% for cefoxitin, 96.4% for cefepime, and 100% for cefalotin, cefotaxime, cefuroxime, cefuroxime axetil, cefixime, ceftriaxone, and ceftazidime. All 28 isolates (100%) were resistant to penicillins and penicillin/β-lactamase inhibitors, including ampicillin, ticarcillin, piperacillin, ampicillin-sulbactam, amoxicillin-clavulanic acid, ticarcillin-clavulanic acid, and piperacillin-tazobactam, and 89.3% of isolates were resistant to aztreonam. Apart from amikacin, resistance rates to other aminoglycosides were high (75% for gentamicin and 82.1% for tobramycin). The quinolone resistance rates were very high, ranging from 64.3% (levofloxacin), 71.4% (ciprofloxacin), 92.9% (ofloxacin, moxifloxacin), to 96.4% (nalidixic acid). Resistance rates to folate pathway inhibitors were also very high (92.9% for trimethoprim-sulfamethoxazole and 96.4% for trimethoprim). Resistance rates to tetracycline and chloramphenicol were 71.4% and 57.1%, respectively. Nevertheless, colistin and amikacin remained active, with susceptibility rates of 96.4% (27/28) and 71.4% (20/28), respectively. Tigecycline showed an average activity of 57.1% (16/28) (Table S1; Fig. 1).

According to the susceptibility profiles observed for all antimicrobials tested, 39.3% (11/28) of the isolates were XDR, whereas 60.7% (17/28) were MDR (Table S1; Fig. S1). Among the 11 XDR CRE isolates, six were *K. pneumoniae*, three *Enterobacter* spp., one *E. coli*, and one *P. mirabilis* (Table S1).

## Carbapenemase and cephalosporinase genes

The prevalence of $bla_{OXA-48}$, $bla_{NDM}$, and $bla_{CMY-1}$ was 53.6% (15/28), 35.7% (10/28), and 25% (7/28), respectively, whereas none of the isolates carried $bla_{KPC}$, $bla_{OXA-23}$, or $bla_{VIM}$ (Fig. S1). Eleven of 17 (64.7%) *K. pneumoniae*, 3/7 (42.9%) *Enterobacter* spp., and 1/3 (33.3%) *E. coli* carried the $bla_{OXA-48}$ gene, whereas 5/17 (29.4%) *K. pneumoniae*, 3/7 (42.9%) *Enterobacter* spp., and 2/3 (66.7%) *E. coli* harbored the $bla_{NDM}$ gene (Table S1; Fig. S1). *E. coli* isolates carried significantly more $bla_{CMY-1}$ than other species ($P = 0.002$). Two of 28 (7.1%) isolates did not carry any of the six carbapenemase genes screened. None of the isolates co-harbored $bla_{OXA-48}$ and $bla_{NDM}$ simultaneously. Nevertheless, 10.7%

**TABLE 1** Prevalence of target broad-spectrum β-lactam resistance genes and clinical sources of *Enterobacterales* isolates in this study

| AMR gene or clinical source of isolates | Prevalence | |
|---|---|---|
| | *n* | % |
| Carbapenemase-encoding genes | | |
| $bla_{OXA-48}$ | 15/28 | 53.6 |
| $bla_{NDM}$ | 10/28 | 35.7 |
| $bla_{KPC}$ | 0/28 | 0 |
| $bla_{VIM}$ | 0/28 | 0 |
| $bla_{OXA-23}$ | 0/28 | 0 |
| Cephalosporinase-encoding gene: $bla_{CMY-1}$ | 7/28 | 25 |
| Sources | | |
| Urine | 16/28 | 57.2 |
| Pus | 7/28 | 25 |
| Blood | 2/28 | 7.1 |
| Bronchial fluid | 2/28 | 7.1 |
| Sputum | 1/28 | 3.6 |

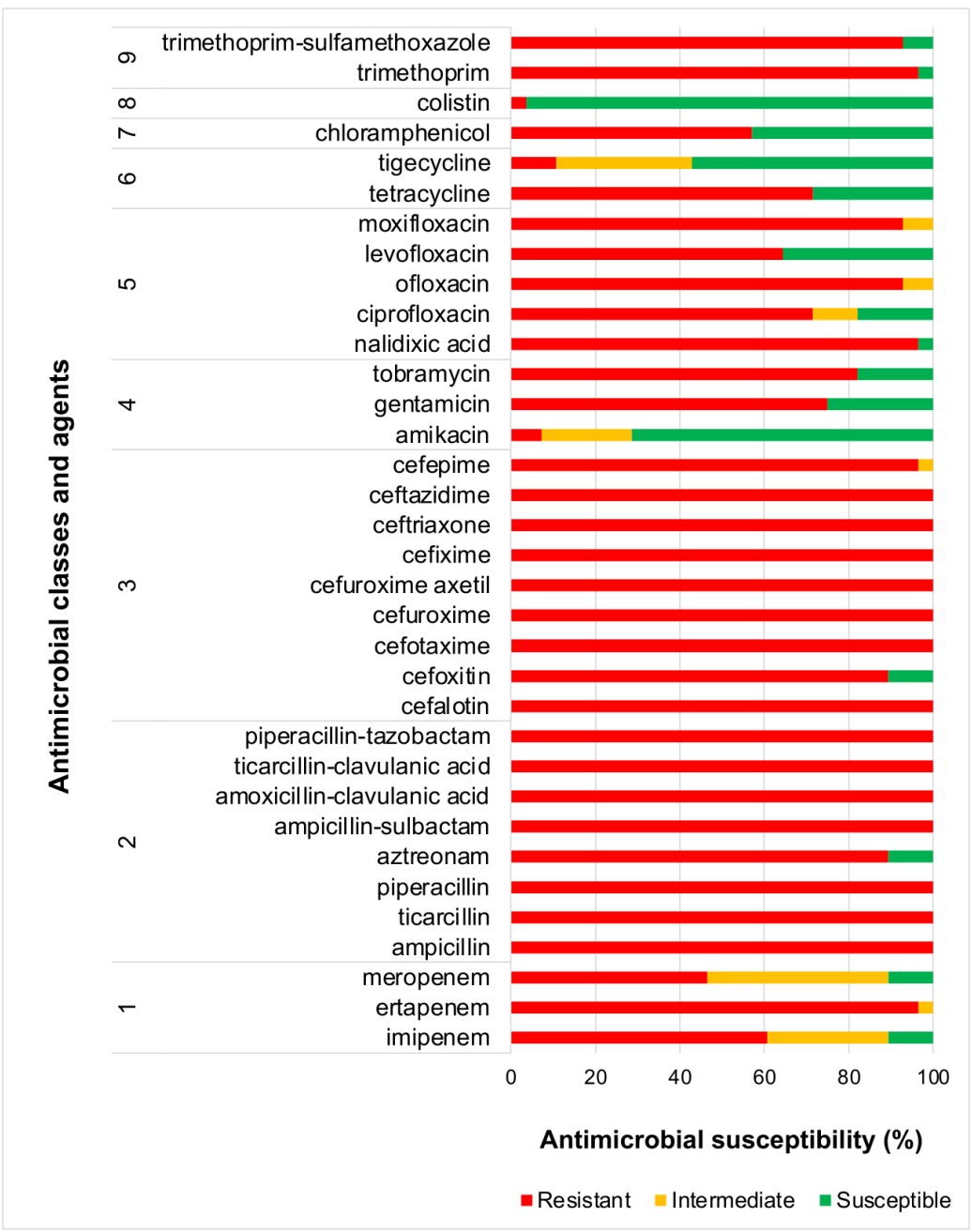

**FIG 1** Overview of antimicrobial resistance profiles of CRE isolates. Antimicrobial classes are as follows: 1: carbapenems; 2: penicillins, penicillins + β-lactamase inhibitors, and monobactams; 3: cephalosporins and cephamycins; 4: aminoglycosides; 5: quinolones; 6: tetracyclines and glycylcyclines; 7: phenicols; 8: polymyxins; and 9: folate pathway inhibitors.

(3/28) isolates carried the combinations ($bla_{NDM}$ + $bla_{CMY-1}$) and ($bla_{OXA-48}$ + $bla_{CMY-1}$) (Table S1).

## Genomic data

Although strains of *K. pneumoniae* and *E. coli* have been identified as XDR and carbapenemase producers, these species are commonly studied for acquired carbapenem resistance. Meanwhile, these resistance traits have been poorly explored in *Entero-bacter* species. Accordingly, two *E. hormaechei* strains, named Eh8322_LBHALD and

Eh202_LBHALD, were identified as XDR and carbapenemase producers and, therefore, were selected for WGS. Both strains were assigned as ST182.

Strain Eh8322_LBHALD (BioSample: SAMN36341330) had one chromosome (GenBank: CP129636) and two plasmids (p1_Eh8322_LBHALD, GenBank: CP129637 and p2_Eh8322_LBHALD, CP129638) with six insertion sequences and 24 horizontal gene transfer (HGT) regions (Table S2). The chromosome of the Eh8322_LBHALD carried multiple resistance genes to antimicrobials, disinfecting agents, antiseptics, and metals, as well as genes associated with virulence (e.g., adherence, invasion, motility, iron uptake, secretion systems, serum resistance, endotoxin secretion, and capsule synthesis regulation) (Table 2). Furthermore, the chromosome harbored two insertion sequences (IS3, IS110) and 19 HGT regions. None of the 19 HGT regions harbored an AMR gene. The p1_Eh8322_LBHALD plasmid belonged to the IncL/M and was 61,054 bp in length. This plasmid carried the $bla_{OXA-48}$ gene, which was embedded between two IS10A elements (IS4-like family transposase genes), forming a composite transposon Tn1999.2 (Fig. 2). The p1_Eh8322_LBHALD had 100% coverage of AP025037 reported in *E. hormachei* (Vietnam), OW849359 in *E. cloacae* (Spain), CP068879 in *K. pneumoniae* (the Netherlands), and CP092010 in *S. enterica* (Switzerland). In addition, CP129637 had 100%, 99.98%, 99.98%, and 99.97% sequence identity with OW849359, AP025037, CP068879, and CP092010, respectively (Fig. 2). The p2_Eh8322_LBHALD plasmid (Fig. 3) belonged to the IncFII(pECLA) and was 91,528 bp in length. This plasmid harbored genes related to biocide and metal resistance as follows: tributyltin resistance (*ygi*W); mercury resistance (*mer*APT); arsenic resistance (*ars*HDA); copper resistance (*pco*E); cation efflux system (*cus*S); and chromium resistance (*srp*C, *chr*B1) (Table 2).

Strain Eh202_LBHALD (BioSample: SAMN35822989) presented one chromosome (GenBank: CP129250), one plasmid (pEh202_LBHALD, GenBank: CP129251), one integron, 16 insertion sequences, and 36 HGT regions. The chromosome of the Eh202_LBHALD strain was similar to that of Eh8322_LBHALD (Table 2). The pEh202_LBHALD plasmid (Fig. 3) was a multireplicon plasmid belonging to the IncFIB(pECLA)/IncFII(pECLA)/IncX3 and was 217,745 bp in length. This plasmid harbored genes related to resistance to β-lactams (*bla*NDM-5, *bla*CTX-M-15, *bla*OXA-1, *bla*TEM-1), fluoroquinolones [*qnr*B17, *aac(6')-Ib-cr*], aminoglycosides [*aac (3)-IIe*, *aph (6)-Id*, *aph(3'')-Ib*], trimethoprim (*dfr*A14), sulfonamides (*sul*2), and tetracycline [*tet*(A)], as well as genes

**TABLE 2** Genomic characteristics of *E. hormaechei* ST182 isolates

| Feature | Eh8322_LBHALD | | | Eh202_LBHALD | |
| --- | --- | --- | --- | --- | --- |
| | Chromosome | p1_Eh8322_LBHALD | p2_Eh8322_LBHALD | Chromosome | pEh202_LBHALD |
| Antimicrobial resistance | $bla_{ACT-20}$, fosA2, PBP3 (D350N, S357N), GyrA (S83I), UhpT (E350Q) | $bla_{OXA-48}$ | $-^a$ | $bla_{ACT-20}$, fosA2, PBP3 (D350N, S357N), GyrA (S83I), UhpT (E350Q) | $bla_{NDM-5}$, $bla_{CTX-M-15}$, $bla_{OXA-1}$, $bla_{TEM-1}$, qnrB17, aac(6')-Ib-cr, aac (3)-IIe, aph (6)-Id, aph(3'')-Ib, dfrA14, sul2, tet(A) |
| Metal tolerance | arsA, arsB, arsC, arsD, copA, cutC,– cusA, cusB, zntA, czcA | | merA, merP, merT, arsA, arsD, arsH, pcoE, cusS, srpC, chrB1, ygiW | arsA, arsB, arsD, copA, cutC, cusA, zntA, czcA | merA, merP, merT, arsA, arsD, arsH, copA, copB, copD, pcoC, pcoE, cusA, cusC, cusB, cusF, silE, silP, ygiW |
| Virulence | mrkC, fimACDFHIZ, csgABCDEFG, hofBC, stgABD, stiB, stkABCDEFG, cheBRWYZ, motAB, iucABCD, iutA, iroN, shuV, chuAS, rfb, htrB, tssI, tssC, clpV, impAF, epsE, flgBCDEFGHIJKLM, flhABCD, fliAEFGHIJLMNPQRSZ, yst1O, exeDF, clpV1, iagB, ehaB, rcsAB | – | – | mrkC, fimACDFHIZ, csgABCDEFG, hofBC, stgABD, stiB, stkABCDEFG, cheBRWYZ, motAB, iucABCD, iutA, iroN, shuV, chuAS, sitD, fepABCDG, fes, rfb, htrB, tssI, tssC, tssD, clpV, impAF, epsE, flgBCDEFGHIJKLM, flhABCD, fliAEFGHIJLMNPQRSZ, yst1O, exeDF, clpV1, iagB, ehaB, rcsAB, sugC, pla | – |
| Plasmid replicon | – | IncL/M | IncFII(pECLA) | – | IncFIB(pECLA)/IncFII(pECLA)/IncX3 |

$^a$–, not found.

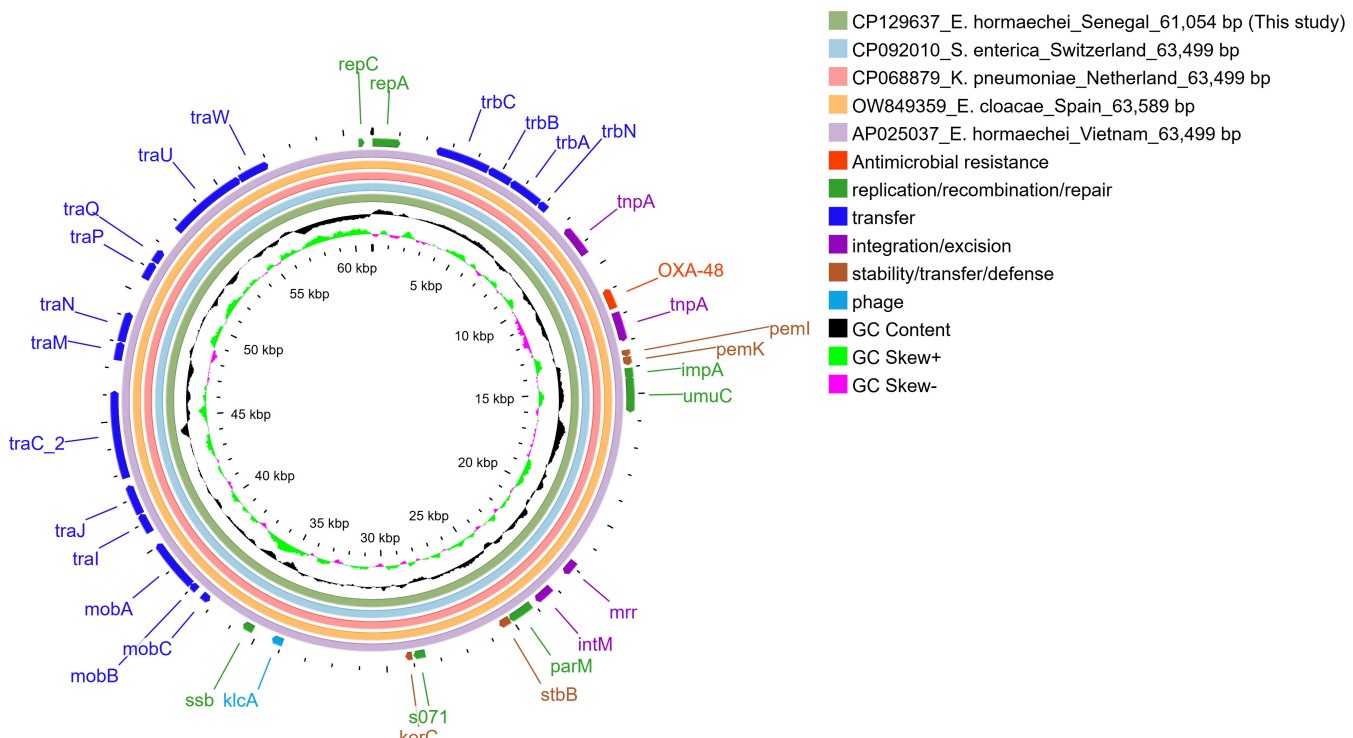

**FIG 2** Circular representation of p1_Eh8322_LBHALD and other closely related plasmids.

associated with biocide and metal resistance, including *ygi*W, *mer*TPA, *ars*HDA, *cop*ABD/*pco*CE (copper resistance), *sil*EP (silver resistance), and *cus*CFBA (cation efflux system) (Table 2). In the NCBI nucleotide database, five plasmids were found to be closely genetically related to pEh202_LBHALD, with SNP pair counts ranging from 0 to 7 and identity ranging from 42.03% to 71.8%) (Fig. 3). Among the five closely related plasmids, p1_Eh8322_LBHALD (from this study) presented 42.03% sequence identity with pEh202_LBHALD and 0 SNP pair count (Fig. 3).

## Phylogenetic analyses

Eh8322_LBHALD exhibited 95.47% identity with Eh202_LBHALD (23 SNP pair counts). Additionally, seven closely related genomes of *E. hormaechei* ST182 strains were found in the NCBI database. All strains were isolated from human samples from Togo (ECLO_616294_SO, GenBank: NRJJ01000000) (98.58% identity with Eh202_LBHALD; 94 SNP pair counts), India (PEER 1096, GenBank: JAQGER010000000) (92.66% identity with Eh202_LBHALD; 451 SNP pair counts), China (EC84, GenBank: JARJGX010000000) (94.73% identity with Eh202_LBHALD; 156 SNP pair counts), Greece (EC-ML559, GenBank: JARUPS010000000) (94.90% identity with Eh202_LBHALD; 559 SNP pair counts), Colombia (44527, GenBank: JZXV01000000) (91.28% identity with Eh202_LBHALD; 499 SNP pair counts), and Myanmar (MY196, GenBank: DAFILT010000000 and M515, GenBank: DACOQQ010000000) (93.63% and 94.95% identity with Eh202_LBHALD; 158 and 151 SNP pair counts). The genome of ECLO_616294_SO, isolated in Togo in 2019, was the most closely related to *E. hormaechei* ST182. Moreover, the phylogenetic tree revealed that Eh202_LBHALD, Eh8322_LBHALD, and ECLO_616294_SO had a recent common ancestor (Fig. S2).

## DISCUSSION

This study offers important insights into the AMR perspective of CRE in a university hospital in Dakar, Senegal. Considering the scarcity of comprehensive research on this

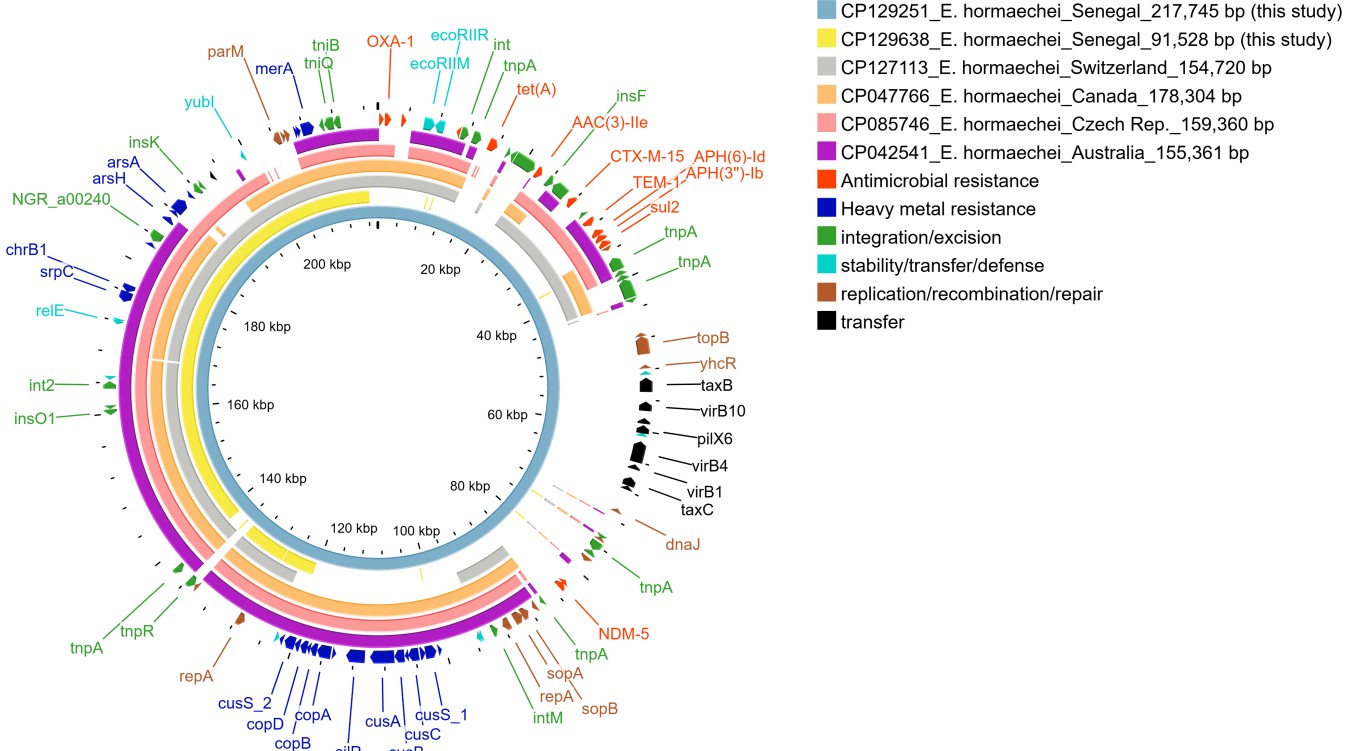

**FIG 3** Circular map of pEh202_LBHALD, p2_Eh8322_LBHALD plasmids, and other closely related plasmids.

area in African countries, these findings establish a crucial foundation for future studies and enhanced surveillance efforts across the region. *K. pneumoniae* was the predominant species, consistent with global trends in CRE epidemiology (22). High resistance rates to commonly used antimicrobials, including penicillins, cephalosporins, and fluoroquinolones, were observed, posing a serious challenge to empirical treatment of community- and hospital-acquired infections.

The high rates of resistance to fluoroquinolones may compromise the empirical treatment of common community-acquired infections, leading to increased treatment failures, prolonged illness, and a greater risk of complications or hospitalizations (23). Additionally, gentamicin resistance in *Enterobacterales* isolated from pediatric sepsis may pose a significant risk of increased morbidity and mortality among children (24). Furthermore, the continued use of these antimicrobial agents in the face of high resistance levels may perpetuate selective pressure, accelerating the spread of MDR strains.

Of clinical importance, the susceptibility rates to colistin and amikacin were notably high, suggesting that these agents may still serve as viable alternatives for treating severe CRE-associated infections. However, their clinical use should be approached with caution. Colistin, for instance, is known for its nephrotoxic effects, which may limit its use in certain patient populations (25). Moreover, increased reliance on colistin has been associated with the spread of mobile colistin resistance *mcr* genes (26). Amikacin, although currently effective against many isolates, is not exempt from the risk of resistance development through enzymatic inactivation or target site modifications (27). Accordingly, tigecycline has been considered an alternative antimicrobial against CRE (28). Unfortunately, only 57.2% of isolates were susceptible to tigecycline in this study. Indeed, tigecycline-resistant *Enterobacterales* strains are becoming widespread, with plasmid-borne *tet*(X) genes encoding tigecycline-inactivating enzymes that have been globally emerging (29). In Senegal, tigecycline prescriptions should be carefully monitored to delay the generalization of tigecycline-resistant bacteria. These concerns

reinforce the need for judicious, stewardship-guided use of the few antimicrobials that still show high susceptibility rates, in order to preserve their clinical effectiveness for as long as possible.

At the molecular level, our findings confirm the dominance of $bla_{OXA-48}$ and $bla_{NDM}$ carbapenemase genes, particularly in *K. pneumoniae* and *Enterobacter* spp. The prevalence of $bla_{OXA-48}$ reported in our study was generally higher than that reported worldwide. In countries across Asia and North Africa, the prevalence of $bla_{OXA-48}$ ranged from 0 to 37.5% (30–34). Although data remain limited due to the scarcity of studies in the country, the occurrence of $bla_{OXA-48}$ in clinical CRE from Dakar has been reported since 2011 (35), suggesting that this gene may have been circulating in the region for over a decade.

The CRE isolates reported in similar studies generally carried more $bla_{NDM}$ than our isolates. Indeed, several studies reported increasing prevalence rates of the $bla_{NDM-1}$ gene, denoting a global dominance trend of this gene (31, 36, 37). Initially confined to the Indian subcontinent (38, 39), the $bla_{NDM}$ gene is now reported globally (40), including in Africa (41–43). Therefore, the detection of $bla_{NDM}$ in *Enterobacterales* from Dakar holds our attention and requires further studies and increased monitoring, particularly because of the broad spectrum of $bla_{NDM}$ hydrolysis and its propagation through plasmids and water networks.

The genomic analysis of two *E. hormaechei* ST182 isolates revealed a high degree of similarity to strains previously reported in West Africa, suggesting regional clonal expansion (41). The genomes from this study were closely related (95.47% of identity and 23 SNP pair counts). Additionally, seven other highly similar genomes of *E. hormaechei* ST182 were identified in the NCBI nucleotide database (identity from 91.28% to 98.58%, and SNP pair counts ranging from 94 to 559). The close phylogenetic relationship among Eh202_LBHALD, Eh8322_LBHALD, and ECLO_616294_SO (a strain isolated in Togo) indicates a recent common ancestor. Therefore, further genomic investigations across the region could shed light on the dissemination of this clonal lineage.

International or intercontinental exchange, possibly via human travelers or migratory birds, may have contributed to the global spread of *E. hormaechei* ST182 (44, 45). The chromosomes of *E. hormaechei* ST182 genomes were highly conserved, sharing almost identical resistomes and virulomes. In contrast, the plasmid content was more variable. This divergence is likely due to rearrangements mediated by mobile genetic elements. Regarding the acquired carbapenem resistance, the $bla_{OXA-48}$ gene was frequently found on IncL/M plasmids. This observation aligns with previous reports indicating that $bla_{OXA-48}$ is commonly located on these plasmid types (46–48). Additionally, pOXA-48 plasmids often carry $bla_{OXA-48}$ as their sole AMR gene (47, 49). These plasmids typically range in size from 60 to 67 kb (47, 49, 50). The plasmid p1_Eh8322_LBHALD displayed 100% coverage and 99.97%–100% sequence identity with pOXA-48 plasmids from CRE in Spain, Vietnam, the Netherlands, and Switzerland, suggesting the widespread circulation of closely related plasmids. The intra- and inter-species dissemination of pOXA-48-like plasmids likely contributes to the global burden of carbapenem resistance.

The *in-silico* characterization of plasmid pEh202_LBHALD, found in a hospital-acquired strain from Dakar, deserves particular attention. This plasmid carries the $bla_{NDM-5}$ gene, which is often associated with rapid plasmid-mediated dissemination of carbapenem resistance, raising concerns over nosocomial outbreaks involving NDM-producing bacteria. Additionally, pEh202_LBHALD encodes resistance determinants to first-line (fluoroquinolones, cephalosporins, aminoglycosides, and sulfonamides) and last-resort (carbapenems) antimicrobials, thereby contributing to the XDR phenotype (Table S1). Besides, the pEh202_LBHALD and p2_Eh8322_LBHALD plasmids harbored tolerance genes to biocides and metals. This feature may facilitate bacterial survival in the hospital environment, even under routine disinfection protocols (51). Thus, further studies are needed to assess the efficacy of hospital cleaning agents against MDR and XDR *Enterobacterales* in Dakar.

Plasmid pEh202_LBHALD carried three replicons [IncFIB(pECLA), IncFII(pECLA), and IncX3], indicating a multireplicon architecture. Indeed, multireplicon plasmids have been identified as harboring antimicrobial resistance genes in *Enterobacterales* (52–54). The fusion of multiple plasmids to form multireplicon elements allows the accumulation of numerous antimicrobial resistance genes, making these plasmids particularly concerning in clinical settings (52). Several virulence genes were identified on the chromosome of *E. hormaechei* ST182 strains. Although *in vivo* expression of these genes was not evaluated in this study, their abundance suggests that these strains may possess a virulent phenotype. Accordingly, our data suggest the presence of virulent, hospital-acquired *E. hormaechei* strains in Dakar that are resistant to nearly all classes of antimicrobials, including carbapenems.

Although not directly evaluated in this study, our findings also raise concerns about the role of suboptimal infection control and hygiene practices in the propagation of antimicrobial-resistant strains within healthcare facilities. Improving hand hygiene, disinfection routines, and hospital wastewater management could be critical interventions to curb the spread of CRE (55).

Despite our findings, some limitations should be acknowledged as follows: (i) only two strains underwent whole-genome sequencing, which restricts the broader interpretation of the genomic data. However, the inclusion of comparative genomes from other regions and the small number of SNP pair counts in the related genomes support the representativeness of these isolates; (ii) this study was limited to a single hospital, and therefore, it may not reflect the nationwide epidemiology of CRE in Senegal; (iii) antimicrobial resistance was phenotypically assessed using an automated system, which may fail to detect certain mechanisms or misclassify borderline resistance profiles; and (iv) although several virulence genes were identified through genomic analysis, *in vivo* virulence assays were not performed, and therefore, the actual pathogenic potential of these strains could not be fully confirmed. Therefore, future studies should include broader, multicenter sampling and integrate complementary phenotypic, molecular, and functional approaches to comprehensively assess CRE dynamics and virulence in West African settings.

## Conclusion

In this study, we report a diversity of carbapenemases among clinical *Enterobacterales* from a university hospital in Dakar, Senegal. We also provide genomic information on *E. hormaechei* ST182 strains harboring $bla_{NDM-5}$ and $bla_{OXA-48}$ genes. *K. pneumoniae* appears to be the main CRE species involved in Dakar, which warrants special attention. Finally, strengthening genomic surveillance in Senegal should be encouraged to map the genomic characteristics and circulation trends of CRE in the country.

## MATERIALS AND METHODS

### Bacterial isolates

This study focused on 28 clinical *Enterobacterales* isolates resistant to at least one carbapenem agent, including imipenem, ertapenem, and meropenem. Isolates were obtained from the biobank of routine activities of Aristide Le Dantec National University Teaching Hospital in 2018 and 2019. The isolates were obtained using eosin methylene blue (EMB) agar (Merck KGaA, Darmstadt, Germany) and identified by Api20E for *Enterobacterales* (bioMérieux, Lyon, France). Bacterial identification was carried out using matrix-assisted laser desorption ionization time-of-flight mass spectrometry (MALDI-TOF MS) (https://maldi-tof-ms-user-platform.ua-bw.de/) (56). Bacterial isolates were stored at −80°C in brain heart infusion broth (Liofilchem S.r.l., Roseto degli Abruzzi, Italy) supplemented with glycerol 15%.

## Antimicrobial susceptibility testing

The VITEK 2 system (bioMérieux, NC, Durham, USA) was used for antimicrobial susceptibility testing (AST). For each isolate, two cards (AST-XN05 and AST-N233) were tested to determine the minimal inhibitory concentration (MIC) according to the manufacturer's instructions. For each isolate, the combination of the AST-XN05 and AST-N233 cards permitted the determination of the MIC of 34 antimicrobials. The MICs were interpreted according to the guidelines of the Clinical and Laboratory Standards Institute (CLSI, 31st edition, M100, 2021). The MDR and XDR profiles were determined according to Magiorakos et al. (57).

## Bacterial DNA extraction

For WGS, genomic DNA (gDNA) was extracted using the MagMAX microbiome ultranucleic acid isolation kit (Applied Biosystems and Thermo Fisher Scientific, Monza, Italy), and the concentration was measured using an Invitrogen Qubit 3 fluorometer (Thermo Fisher Scientific Inc., Strasbourg, France). For end-point PCR, the mechanical thermal lysis method was used to extract the bacterial gDNA. Briefly, a bacterial colony was dispersed in a tube containing 1 mL of sterile distilled water, vortexed, boiled for 15 min at 100℃, and centrifuged at 13,200 rpm for 10 min. The supernatant was carefully recovered, aliquoted, and stored at −20℃.

## Amplification of carbapenemase genes

Each gDNA sample was subjected to simplex end-point PCR using an Applied Biosystems 2720 thermal cycler (Applied Biosystems, Foster City, CA, USA). Specific primers (Table S3) were used to amplify carbapenemase genes ($bla_{NDM}$, $bla_{KPC}$, $bla_{OXA-48}$, $bla_{OXA-23}$, and $bla_{VIM}$), and the plasmid-mediated cephalosporinase gene $bla_{CMY-1}$. Each reaction included both positive and negative controls. The final volume for each PCR was 20 μL (2.5 μL of DNA + 17.5 μL mix). The mix was prepared using FIREPol Master Mix (Solis BioDyne Teaduspargi 9, 50411 Tartu, Estonia). Gene amplification was performed using the following program: initial denaturation at 95℃ for 3 min, 35 PCR cycles (denaturation: 94℃ for 30 s; hybridization; extension: 72℃ for 60 s), and a final elongation at 72℃ for 7 min. Each amplicon (10 μL) was separated on a 2% agarose gel in 1× TAE buffer for 35 min at 135 volts, and the amplified fragment was detected using a GelDoc imager (BioRad, Hercules, CA, USA).

## Whole-genome sequencing and genomic analyses

Isolates Eh8322_LBHALD (BioSample: SAMN36341330) and Eh202_LBHALD (BioSample: SAMN35822989) were submitted to WGS on an Illumina MiSeq platform using v2 reagent kits, generating 2 × 150 bp paired-end reads (Illumina, San Diego, CA, USA). The quality of the raw data has been evaluated with FastQC v.0.11.90 (https://github.com/s-andrews/FastQC/releases/tag/v0.11.9) before cleaning using PRINSEQ v.0.20.4 (58) and Cutadapt 3.4 (59). The reads were *de novo* assembled using SPAdes v.3.15.4 (60). Contigs of less than 200 bp (which did not contain usable information owing to their small size) were removed from further analysis. Assembly quality was evaluated using Bowtie2 v.2.4.4 (61), Samtools v.2.02 (62), and BUSCO v.5.4.4 (63). To identify any possible DNA contamination, the assembly obtained in the previous step was submitted to the RefSeq database (64). Structural annotation was performed using Prokka v.1.14.5 (65) and its dependencies (UniProt KB database, NCBI Bacterial Antimicrobial Resistance Reference Gene Database (https://www.ncbi.nlm.nih.gov/bioproject/313047), Barrnap v.0.9 (https://github.com/tseemann/barrnap), Prodigal v.2.6.3 (66), and the EggNOG v.5.0 database (67). The prediction of 16S rRNA was conducted using Barrnap v.0.9. ARGs were searched using Prodigal and the Comprehensive Antibiotic Resistance Database (CARD) [(68); https://card.mcmaster.ca/]. Virulence genes were identified using the Virulence Factor Database (VFDB) (69).

The plasmids were assembled from the WGS FASTQ data using PLACNETw [https://castillo.dicom.unican.es/upload/; (70)] and annotated using Prokka v.1.14.5 and CARD. Circular plasmid representations were generated using PROKSEE [(71); https://proksee.ca/projects]. The insertion sequences and transposons were identified using ISEScan version v.1.7.2.3 (72) and MobileElementFinder v1.0.3 (73). The horizontal gene transfer (HGT) regions were identified using Alien-hunter v.1.7 (74). The integrons were detected using IntegronFinder v.2.0 (75).

Genomic sequences closely related to those found in this study were identified using the Basic Local Alignment Search Tool (BLAST) (https://blast.ncbi.nlm.nih.gov/Blast.cgi). Additionally, sequence alignments were conducted using CSIPhylogeny v.1.4 (76), and the same tool provided the number of single-nucleotide polymorphisms (SNPs) and core genome identity percentages. Moreover, a phylogenetic tree was generated using the interactive tree of life (iTOL) v.6.1 (77).

## Statistical analysis

Microsoft Excel was used for data analysis. $\chi^2$ at 5% risk was used, and $P$-values were obtained from the proportion comparison test. Statistical significance was set at $P < 0.05$.

## ACKNOWLEDGMENTS

We are grateful to all members of the Pole of Microbiology of the Institut Pasteur de Dakar, rime bioinformatics, and the bacteriology-virology laboratory of the national university hospital Aristide Le Dantec of Dakar. J.P.R.F. is a research fellow at the São Paulo Research Foundation (FAPESP) (grant number: 23/16216-4).

K.M.D.: conceptualization, methodology, investigation, data curation, formal analysis, software, and writing-original draft. B.S.B.: conceptualization, methodology, supervision, writing – review & editing. G.L.: project administration, supervision, resources. F.P.S.: data curation, formal analysis, software, and writing-original draft. J.P.R.F.: data curation, formal analysis, software, and writing-original draft. A.C.: data curation, formal analysis, software, writing – review & editing. G.A.: data curation, formal analysis, software, writing – review & editing. A.B.G.: data curation, formal analysis, software. A.B.-D.: project administration, resources. A.D.: resources, investigation. F.P.L.: resources, investigation. A.C.: resources, investigation. S.M.L.N.: resources, investigation. A.T.: resources, investigation. F.K.: resources, investigation. H.D.-S.: resources, investigation. S.N.-C.: resources, investigation. H.D.-N.: project administration, resources. C.T.-K.: project administration, resources. A.G.-D.: project administration, resources. S.D.: supervision, writing – review & editing. S.M.: project administration, resources. C.S.B.B.: project administration, resources. A.S.: project administration, resources, supervision. M.C.: conceptualization, methodology, resources, validation, supervision, writing – review & editing.

## AUTHOR AFFILIATIONS

[1]Bacteriology-Virology laboratory, National University Hospital Aristide Le Dantec, Dakar, Senegal
[2]Department of Microbiology, Global Health Research Institute, Lomé, Togo
[3]WHO Country Office Senegal, World Health Organization, Dakar, Senegal
[4]Institut de Recherche en Santé, de Surveillance Epidémiologique et de Formation (IRESSEF), Dakar, Senegal
[5]Department of Internal Medicine, School of Veterinary Medicine and Animal Science, University of São Paulo, São Paulo, Brazil
[6]School of Veterinary Medicine, Metropolitan University of Santos, Santos, Brazil
[7]Paulista School of Medicine, Federal University of São Paulo, São Paulo, Brazil
[8]Rime Bioinformatics, Palaiseau, Île-de-France, France
[9]Pole of Microbiology, Institut Pasteur de Dakar, Dakar, Senegal
[10]Faculté des Sciences de la Santé, Université de Kara, Kara, Togo
[11]Medical Analysis laboratory, Institut Pasteur de Dakar, Dakar, Senegal

## AUTHOR ORCIDs

Komla Mawunyo Dossouvi http://orcid.org/0000-0003-0967-5314
Fábio Parra Sellera http://orcid.org/0000-0002-4725-0125
João Pedro Rueda Furlan http://orcid.org/0000-0003-2516-1129

## AUTHOR CONTRIBUTIONS

Komla Mawunyo Dossouvi, Conceptualization, Data curation, Formal analysis, Investigation, Methodology, Software, Writing – original draft | Bissoume Sambe Ba, Conceptualization, Methodology, Supervision, Writing – review and editing | Gora Lo, Project administration, Resources, Supervision | Fábio Parra Sellera, Data curation, Formal analysis, Software, Writing – original draft, Writing – review and editing | João Pedro Rueda Furlan, Data curation, Formal analysis, Software, Writing – original draft, Writing – review and editing | Antoine Culot, Data curation, Formal analysis, Software, Writing – review and editing | Guillaume Abriat, Data curation, Formal analysis, Software, Writing – review and editing | Adja Bousso Gueye, Data curation, Formal analysis, Software | Awa Ba-Diallo, Project administration, Resources | Assane Dieng, Investigation, Resources | Fatime Poulo Ly, Investigation, Resources | Abdoulaye Cissé, Investigation, Resources | Serigne Mbaye Lo Ndiaye, Investigation, Resources | Alioune Tine, Investigation, Resources | Farba Karam, Investigation, Resources | Habsa Diagne-Samb, Investigation, Resources | Safietou Ngom-Cisse, Investigation, Resources | Halimatou Diop-Ndiaye, Project administration, Resources | Coumba Toure-Kane, Project administration, Resources | Aïssatou Gaye-Diallo, Project administration, Resources | Sika Dossim, Supervision, Writing – review and editing | Souleymane Mboup, Project administration, Resources | Cheikh Saad Bouh Boye, Project administration, Resources | Abdoulaye Seck, Project administration, Resources, Supervision | Makhtar Camara, Conceptualization, Methodology, Resources, Supervision, Validation, Writing – review and editing

## DATA AVAILABILITY

The authors confirm that the data supporting the findings of this study are available in the article and/or its supplemental materials. The genomic data of all samples were deposited in the NCBI database under BioProjects PRJNA986271 and PRJNA992103.

## ETHICS APPROVAL

This study has received the ethical research approval of the Research Ethics Committee (CER) of Cheikh Anta Diop University (UCAD) under the reference CER/UCAD/AD/MSN/051/2020.

## ADDITIONAL FILES

The following material is available online.

### Supplemental Material

**Supplemental Material (Spectrum00780-25-s0001.docx).** Tables S1 to S3; Fig. S1 and S2.

### Open Peer Review

**PEER REVIEW HISTORY (review-history.pdf).** An accounting of the reviewer comments and feedback.

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
