## [Reviewer comments · Microbiology Spectrum]

Microbiology Spectrum

Clinical carbapenem-resistant *Enterobacterales* in a university hospital in Dakar, Senegal: genomic insights into *Enterobacter hormaechei* ST182 strains carrying *bla*_{NDM-5} and *bla*_{OXA-48} genes

Komla Dossouvi, Bissoume Sambe Ba, Gora Lô, Fábio Sellera, João Pedro Furlan, Antoine Culot, Guillaume Abriat, Adja Gueye, Awa Ba-Diallo, Assane Dieng, Fatime Ly, Abdoulaye Cissé, Serigne Ndiaye, Alioune Tine, Farba Karam, Habsa Diagne-Samb, Safietou Ngom-Cisse, Halimatou Diop-Ndiaye, Coumba Toure-Kane, Aïssatou Gaye-Diallo, Sika Dossim, Souleymane Mboup, Cheikh Boye, Abdoulaye Seck, and Makhtar Camara

Corresponding Author(s): Komla Dossouvi, Hopital Aristide Le Dantec

Review Timeline:

Submission Date:	March 17, 2025
Editorial Decision:	June 19, 2025
Revision Received:	July 29, 2025
Accepted:	August 18, 2025

Editor: Felix Toka

Reviewer(s): Disclosure of reviewer identity is with reference to reviewer comments included in decision letter(s). The following individuals involved in review of your submission have agreed to reveal their identity: Beiwen Zheng (Reviewer #1)

Transaction Report:

DOI: <https://doi.org/10.1128/spectrum.00780-25>

Re: Spectrum00780-25 (**Prevalence of clinical carbapenem-resistant *Enterobacterales* in Dakar, Senegal: genomic insights into *Enterobacter hormaechei* ST182 carrying *bla*_{NDM-5} and *bla*_{OXA-48} genes**)

Dear Dr. Komla Mawunyo Dossouvi:

Thank you for the privilege of reviewing your work. Below you will find my comments, instructions from the Spectrum editorial office, and the reviewer comments.

Your manuscript offers a valuable contribution to the understanding of carbapenem-resistant Enterobacterales (CRE) in sub-Saharan Africa, particularly through the use of whole-genome sequencing (WGS) and plasmid analysis. To further strengthen the manuscript, please address Reviewers comments and suggestions.

Revision Guidelines

Sincerely,
Felix Toka
Editor
Microbiology Spectrum

Reviewer #1 (Comments for the Author):

General comments :

1. The manuscript addresses a significant global public health issue-carbapenem-resistant Enterobacterales (CRE)-in the

context of sub-Saharan Africa, specifically Dakar, Senegal. Given the limited genomic data from this region, this study significantly contributes to the field, enhancing regional and global understanding of CRE epidemiology.

2. The combination of phenotypic antimicrobial susceptibility testing, molecular detection of carbapenemase genes, and whole-genome sequencing (WGS) provides comprehensive insights. Particularly commendable is the detailed plasmid analysis, providing valuable data on the genetic context and potential for horizontal gene transfer. The manuscript effectively employs bioinformatics tools to elucidate resistome, virulome, and plasmid content. The identification of plasmids carrying critical resistance determinants (*bla*NDM-5, *bla*OXA-48) adds substantial value. However, more detailed exploration into the genomic mechanisms driving resistance gene dissemination would enhance the manuscript.

3. The identification of amikacin and colistin as potentially viable therapeutic alternatives is clinically valuable, particularly in resource-limited settings. Nevertheless, further discussion on the potential limitations or consequences (e.g., nephrotoxicity associated with colistin or potential resistance emergence) could enrich clinical implications.

Minor comments

1. Clarify the abbreviation "WGSBA" used in the abstract; consider using standard terminology such as "WGS analysis."

2. Include brief context regarding the clinical implications of CRE in resource-limited settings early in the introduction to clearly establish the relevance of the study.

3. Provide more details on patient demographics or clinical settings (e.g., wards, outpatient vs. inpatient) from which isolates were obtained to enhance clinical context.

4. Indicate clearly if ethical approval or exemption was obtained for utilizing clinical isolates from routine diagnostics.

5. Clarify the rationale behind selecting only two isolates for detailed genomic analysis. If these represent unique phenotypes or genotypes, explicitly state this in the results.

6. Consider presenting data on patient outcomes or infection types (if available), to correlate resistance profiles with clinical impact.

7. Expand the discussion of the clinical and public health implications of resistance to first-line agents like fluoroquinolones and gentamicin, particularly in relation to local prescribing practices.

8. Elaborate briefly on the potential role of hospital hygiene and infection control measures in the dissemination of these extensively drug-resistant (XDR) and multidrug-resistant (MDR) organisms in healthcare settings.

9. The figures (plasmid comparisons) are clear but adding annotations directly within the figures, particularly for key resistance genes and mobile genetic elements, would further enhance readability and interpretability.

10. Supplementary tables provide extensive detail, but the primary manuscript could benefit from a brief summary table highlighting key resistance genes, plasmid types, and clinical sources of isolates.

Reviewer #2 (Comments for the Author):

This manuscript entitled "Prevalence of clinical carbapenem-resistant Enterobacterales in Dakar, Senegal: genomic insights into *Enterobacter hormaechei* ST182 carrying *bla*NDM-5 and *bla*OXA-48 genes", briefly described the 2018-2019 prevalence of CRE in a hospital in Dakar, and two of the *Enterobacter hormaechei* strains were whole genome sequenced and analyzed. I can understand that this study did not allow for more in-depth analysis due to laboratory and funding constraints. But using data from 28 CREs in one hospital to describe the prevalence of Dakar is not accurate and was suspected of being exaggerated. And the conflict between ESBL phenotypes and carbapenemases was confusing to me. And the authors did not emphasize the uniqueness of the two *Enterobacter hormaechei* strains. Authors should double-check the results and revise the manuscript, streamlining the language and not going on at length. Figures also need to be embellished.

Major comments :

1. Line 294-296 All 28 strains were obtained from the Aristide le Dantec National University Teaching Hospital. The number of strains is too small and from a single source, I don't think this is representative of the prevalence of CRE in Dakar, so the title needs to be changed. Please avoid exaggeration in the title.

2. How many Enterobacterales isolates were isolated during this study and what was the isolation rate of CRE?

3. I don't think the presentation on carbapenemase genes in TableS1 is a good format, and I suggest changing Fig S1 to a stacked bar chart to add relevant species information.

4. Line 121-122 Strains with no detectable carbapenemase genes should be performed further CIM (carbapenem inactivation method) experiments.

5. Is the ESBL phenotype associated with carrying *bla*CMY-1? Why do strains carrying *bla*NDM or *bla*OXA-48 develop an ESBL phenotype? The ESBL phenotype is generally masked by carbapenemases. Vietek's results may not be accurate and the authors are advised to refer to CLSI to reconfirm the ESBL phenotype.

6. Line 113-114 and Line 127-128 There are 11 XDR strains, why were only two *E. hormaechei* strains selected for genome sequencing and what were the principles of selection? What was the purpose of selecting these strains for sequencing?

7. Line 170-180 The ANI is often used to measure whether or not it is the same genus. SNPs are recommended to measure relatedness between strains. The authors should further calculate the number of SNPs in the core genome between ST182 strains.

8. The discussion section is too lengthy, and the author narrates everything relevant in the results section in the discussion. I was unable to grasp what the central point was in the author's discussion. The authors also did not state the shortcomings and

limitations of this study in the discussion section.

Minor comments :

1. Line 104-109 The authors' interpretation of the drug sensitivity results was incomplete, and the 34 drugs were suggested to be described in order according to drug classification (e.g., carbapenems, cephalosporins, quinolones, etc.). It is also recommended that the drugs in Fig 1 be similarly categorised.
2. Line 121 "p" should be italicized.
3. Fig 2B contains the contents of Fig 2A, so Fig 2A is unnecessary. The same is true for Fig 4A and Fig 4B. Fig 2A and Fig 4A should be deleted.
4. Line 152-156 The two strains have similar chromosomes and do not need to be described separately, which is too wordy.

This manuscript entitled “Prevalence of clinical carbapenem-resistant Enterobacterales in Dakar, Senegal: genomic insights into *Enterobacter hormaechei* ST182 carrying blaNDM-5 and blaOXA-48 genes”, briefly described the 2018-2019 prevalence of CRE in a hospital in Dakar, and two of the *Enterobacter hormaechei* strains were whole genome sequenced and analyzed. I can understand that this study did not allow for more in-depth analysis due to laboratory and funding constraints. But using data from 28 CREs in one hospital to describe the prevalence of Dakar is not accurate and was suspected of being exaggerated. And the conflict between ESBL phenotypes and carbapenemases was confusing to me. And the authors did not emphasize the uniqueness of the two *Enterobacter hormaechei* strains. Authors should double-check the results and revise the manuscript, streamlining the language and not going on at length. Figures also need to be embellished.

Major comments :

1. Line 294-296 All 28 strains were obtained from the Aristide le Dantec National University Teaching Hospital. The number of strains is too small and from a single source, I don't think this is representative of the prevalence of CRE in Dakar, so the title needs to be changed. Please avoid exaggeration in the title.
2. How many *Enterobacterales* isolates were isolated during this study and what was the isolation rate of CRE?
3. I don't think the presentation on carbapenemase genes in TableS1 is a good format, and I suggest changing Fig S1 to a stacked bar chart to add relevant species information.
4. Line 121-122 Strains with no detectable carbapenemase genes should be performed further CIM (carbapenem inactivation method) experiments.
5. Is the ESBL phenotype associated with carrying blaCMY-1? Why do strains carrying blaNDM or blaOXA-48 develop an ESBL phenotype? The ESBL phenotype is generally masked by carbapenemases. Vietek's results may not be accurate and the authors are advised to refer to CLSI to reconfirm the ESBL phenotype.
6. Line 113-114 and Line 127-128 There are 11 XDR strains, why were only two *E. hormaechei* strains selected for genome sequencing and what were the principles of selection? What was the purpose of selecting these strains for sequencing?
7. Line 170-180 The ANI is often used to measure whether or not it is the same genus. SNPs are recommended to measure relatedness between strains. The authors should further calculate the number of SNPs in the core genome between ST182 strains.
8. The discussion section is too lengthy, and the author narrates everything relevant in the results section in the discussion. I was unable to grasp what the central point was in the author's discussion. The authors also did not state the shortcomings and limitations of this study in the discussion section.

Minor comments :

1. Line104-109 The authors' interpretation of the drug sensitivity results was incomplete, and the 34 drugs were suggested to be described in order according to drug classification (e.g., carbapenems, cephalosporins, quinolones, etc.). It is also recommended that the drugs in Fig1

be similarly categorised.

2. Line 121 “p” should be italicized.
3. Fig 2B contains the contents of Fig 2A, so Fig 2A is unnecessary. The same is true for Fig 4A and Fig 4B. Fig 2A and Fig 4A should be deleted.
4. Line 152-156 The two strains have similar chromosomes and do not need to be described separately, which is too wordy.

Response to Reviewers

Dear Reviewer #1,

Thank you very much for your time and consideration of our manuscript. Below you will find our responses for your comments. All changes are highlighted in yellow in the manuscript.

Reviewer #1:

General comments:

1. The manuscript addresses a significant global public health issue-carbapenem-resistant *Enterobacteriales* (CRE)-in the context of sub-Saharan Africa, specifically Dakar, Senegal. Given the limited genomic data from this region, this study significantly contributes to the field, enhancing regional and global understanding of CRE epidemiology.

R. Thank you very much for your comment.

2. The combination of phenotypic antimicrobial susceptibility testing, molecular detection of carbapenemase genes, and whole-genome sequencing (WGS) provides comprehensive insights. Particularly commendable is the detailed plasmid analysis, providing valuable data on the genetic context and potential for horizontal gene transfer. The manuscript effectively employs bioinformatics tools to elucidate resistome, virulome, and plasmid content. The identification of plasmids carrying critical resistance determinants (*bla_{NDM-5}*, *bla_{OXA-48}*) adds substantial value. However, more detailed exploration into the genomic mechanisms driving resistance gene dissemination would enhance the manuscript.

*R. The *bla_{OXA-48}* and *bla_{NDM-5}* genes were identified on single and multireplicon plasmids, respectively. The plasmid families found are known to spread antimicrobial resistance genes worldwide. In this regard, the role of these plasmids was enhanced in the manuscript as suggested.*

*Lines 259-263: Regarding the acquired carbapenem resistance, the *bla_{OXA-48}* gene was frequently found on IncL/M plasmids. This observation aligns with previous reports indicating that *bla_{OXA-48}* is commonly located on these plasmid types (46–48). Additionally, pOXA-48 plasmids often carry *bla_{OXA-48}* as their sole AMR gene (47, 49). These plasmids typically range in size from 60 to 67 kb (47, 49, 50).*

Lines 279-284: Plasmid pEh202_LBHALD carried three replicons [IncFIB(pECLA), IncFII(pECLA), and IncX3], indicating a multireplicon architecture. Indeed, multireplicon plasmids have been identified in as harboring antimicrobial resistance genes in Enterobacterales (52–54). The fusion of multiple plasmids to form multireplicon elements allows the accumulation of numerous antimicrobial resistance genes, making these plasmids particularly concerning in clinical settings (52).

3. The identification of amikacin and colistin as potentially viable therapeutic alternatives is clinically valuable, particularly in resource-limited settings. Nevertheless, further discussion on the potential limitations or consequences (e.g., nephrotoxicity associated with colistin or potential resistance emergence) could enrich clinical implications.

R. Thank you for this suggestion. The following sentence was included in the Discussion section to address this issue: “Of clinical importance, the susceptibility rates to colistin and amikacin were notably high, suggesting that these agents may still serve as viable alternatives for treating severe CRE-associated infections. However, their clinical use should be approached with caution. Colistin, for instance, is known for its nephrotoxic effects, which may limit its use in certain patient populations (25). Moreover, increased reliance on colistin has been associated with the spread of mobile colistin resistance mcr genes (26). Amikacin, while currently effective against many isolates, is not exempt from the risk of resistance development through enzymatic inactivation or target site modifications (27) (Lines 217-224).

Minor comments

1. Clarify the abbreviation "WGSBA" used in the abstract; consider using standard terminology such as "WGS analysis."

R. Thank you for the suggestion. We have replaced the non-standard abbreviation "WGSBA" with the more widely accepted term "WGS analysis" in both the abstract and the main text (Lines 33-34, 39 and 96).

2. Include brief context regarding the clinical implications of CRE in resource-limited settings early in the introduction to clearly establish the relevance of the study.

R. Thank you for this suggestion. We have included the following sentence to address this issue: “In resource-limited settings, the clinical impact of CRE is particularly more critical due to the restricted availability of advanced diagnostic tools, limited treatment options, and inadequate infection control infrastructure (9). These challenges increase the risk of therapeutic failure, prolonged hospital stays, and mortality, making CRE a critical concern in such regions (10)” (Lines 72-76).

3. Provide more details on patient demographics or clinical settings (e.g., wards, outpatient vs. inpatient) from which isolates were obtained to enhance clinical context.

R. We have provided more details such as wards and outpatient vs. inpatient data in the manuscript (Lines 103-107).

4. Indicate clearly if ethical approval or exemption was obtained for utilizing clinical isolates from routine diagnostics.

R. We have included the ethical approval of our study (Lines 393-396).

5. Clarify the rationale behind selecting only two isolates for detailed genomic analysis. If these represent unique phenotypes or genotypes, explicitly state this in the results.

R. We have explained why we selected only two isolates for detailed genomic analysis in the manuscript (Lines 353-355). The main reason for this was the lack of funding to subcontract the WGS analysis for all the 28 isolates.

Lines 144-148: Although strains of K. pneumoniae and E. coli have been identified as XDR and carbapenemase producers, these species are commonly studied for acquired carbapenem resistance. Meanwhile, these resistance traits have been little explored in Enterobacter species. Accordingly, two E. hormaechei strains, named Eh8322_LBHALD and Eh202_LBHALD, were identified as XDR and carbapenemase producers and, therefore, were selected for WGS.

6. Consider presenting data on patient outcomes or infection types (if available), to correlate resistance profiles with clinical impact.

R. We added a summary of clinical sources of isolates in Table 1. However, data on the patient outcomes were not available. We acknowledge that the inclusion of patient outcomes in the manuscript could have improved the manuscript.

7. Expand the discussion of the clinical and public health implications of resistance to first-line agents like fluoroquinolones and gentamicin, particularly in relation to local prescribing practices.

R. Thank you for this suggestion. We have included the following sentence to address this issue: “The high rates of resistance to fluoroquinolones may compromise the empirical treatment of common community-acquired infections, leading to increased treatment failures, prolonged illness, and a greater risk of complications or hospitalizations (23). Additionally, gentamicin resistance in Enterobacterales isolated from pediatric sepsis may pose a significant risk of increased morbidity and mortality among children (24). Furthermore, the continued use of these antimicrobial agents in the face of high resistance levels may perpetuate selective pressure, accelerating the spread of MDR strains.” (Lines 210-216)

8. Elaborate briefly on the potential role of hospital hygiene and infection control measures in the dissemination of these extensively drug-resistant (XDR) and multidrug-resistant (MDR) organisms in healthcare settings.

R. We have added the potential role of hospital hygiene and infection control measures in the dissemination of XDR and MDR organisms in healthcare settings, and provided recommendations. Lines 289-292: “While not directly evaluated in this study, our findings also raise concerns about the role of suboptimal infection control and hygiene practices in the propagation of antimicrobial-resistant strains within healthcare facilities. Improving hand hygiene, disinfection routines, and hospital wastewater management could be critical interventions to curb the spread of CRE (55).”

9. The figures (plasmid comparisons) are clear but adding annotations directly within the figures, particularly for key resistance genes and mobile genetic elements, would further enhance readability and interpretability.

R. Thank you for your remark.

10. Supplementary tables provide extensive detail, but the primary manuscript could benefit from a brief summary table highlighting key resistance genes, plasmid types, and clinical sources of isolates.

R. We have included in the primary manuscript a brief summary table highlighting key resistance genes, plasmid types, and clinical sources of isolates. Table 1 and 2.

Dear Reviewer #2,

Thank you very much for your time and consideration of our manuscript. Below you will find our responses for your comments. All changes are highlighted in yellow in the manuscript.

Reviewer #2:

This manuscript entitled "Prevalence of clinical carbapenem-resistant *Enterobacterales* in Dakar, Senegal: genomic insights into *Enterobacter hormaechei* ST182 carrying *bla*_{NDM-5} and *bla*_{OXA-48} genes", briefly described the 2018-2019 prevalence of CRE in a hospital in Dakar, and two of the *Enterobacter hormaechei* strains were whole genome sequenced and analyzed. I can understand that this study did not allow for more in-depth analysis due to laboratory and funding constraints. But using data from 28 CREs in one hospital to describe the prevalence of Dakar is not accurate and was suspected of being exaggerated. And the conflict between ESBL phenotypes and carbapenemases was confusing to me. And the authors did not emphasize the uniqueness of the two *Enterobacter hormaechei* strains. Authors should double-check the results and revise the manuscript, streamlining the language and not going on at length. Figures also need to be embellished.

*R. We appreciate the reviewer's thoughtful comments and acknowledge the valid concerns raised. In response, we have revised the manuscript title to more accurately reflect the scope of our study and avoid overgeneralization regarding the prevalence of CRE in Dakar. Additionally, we removed the mention of ESBL co-detection in carbapenemase-producing *Enterobacterales*, as we recognize that carbapenemase activity typically interferes with the detection of ESBL phenotypes (e.g., ghost zone formation). We thank the reviewer for pointing this out. We have also expanded the description of the two sequenced *Enterobacter hormaechei**

strains to better highlight their genomic features and epidemiological relevance. The figures have been revised and improved according to the reviewers' suggestions to enhance clarity and presentation. We hope these changes address the reviewer's concerns and contribute to the overall quality of the manuscript.

Major comments :

1. Line 294-296 All 28 strains were obtained from the Aristide le Dantec National University Teaching Hospital. The number of strains is too small and from a single source, I don't think this is representative of the prevalence of CRE in Dakar, so the title needs to be changed. Please avoid exaggeration in the title.

R. We changed the title. Now its reads: "Clinical carbapenem-resistant Enterobacterales in a university hospital in Dakar, Senegal: genomic insights into Enterobacter hormaechei ST182 strains carrying bla_{NDM-5} and bla_{OXA-48} genes".

2. How many *Enterobacterales* isolates were isolated during this study and what was the isolation rate of CRE?

R. The isolates used in this study were retrieved from the laboratory's biobank, and the total number of Enterobacterales strains isolated between 2018 and 2019 is not available. Consequently, we removed the term "prevalence" from the title and specified the use of biobank samples in the Materials and Methods section (Lines 1–3 and 317).

3. I don't think the presentation on carbapenemase genes in Table S1 is a good format, and I suggest changing Fig. S1 to a stacked bar chart to add relevant species information.

R. We have changed the presentation on carbapenemase genes in Table S1. Now the presence of AMR genes is highlighted in gray in the Table S1. In addition, we have changed Fig S1 to a stacked bar chart.

4. Line 121-122 Strains with no detectable carbapenemase genes should be performed further CIM (carbapenem inactivation method) experiments.

R. Thank you for your valuable suggestion. We agree that performing additional phenotypic tests such as the CIM would help to clarify the resistance mechanisms in strains with no

detectable carbapenemase genes. Unfortunately, due to funding limitations, it was not feasible to conduct these experiments during the current study. However, we consider this an important point and plan to include such analyses in future investigations as resources allow.

5. Is the ESBL phenotype associated with carrying *bla*_{CMY-1}? Why do strains carrying *bla*_{NDM} or *bla*_{OXA-48} develop an ESBL phenotype? The ESBL phenotype is generally masked by carbapenemases. Vietek's results may not be accurate and the authors are advised to refer to CLSI to reconfirm the ESBL phenotype.

R. We thank the reviewer for this important observation and agree with the comment. Indeed, the ESBL phenotype is generally masked by the presence of carbapenemases, and the co-detection as originally presented may not be accurate. In light of this, we have removed the data referring to the ESBL phenotype in carbapenemase-producing strains from the manuscript.

6. Line 113-114 and Line 127-128 There are 11 XDR strains, why were only two *E. hormaechei* strains selected for genome sequencing and what were the principles of selection? What was the purpose of selecting these strains for sequencing?

*R. Thank you for these insightful comments. Our initial aim was to perform whole-genome sequencing (WGS) on all 28 CRE isolates; however, we encountered two main limitations: the lack of local sequencing infrastructure and restricted funding for outsourcing WGS. Due to these constraints, we self-financed the sequencing of two isolates as a starting point to generate preliminary genomic data. These data were intended to guide future research directions and support applications for external funding and collaborations to enable the sequencing of the remaining isolates. We chose two *E. hormaechei* strains for WGS because this specie is a clinically relevant member of the ESKAPE group, often associated with multidrug resistance. Despite its importance, carbapenem resistance mechanisms in *Enterobacter* spp. remain underexplored compared to other ESKAPE pathogens. We have added a clarifying sentence in the revised manuscript to explain the rationale behind this selection.*

*Lines 144-148: Although strains of *K. pneumoniae* and *E. coli* have been identified as XDR and carbapenemase producers, these species are commonly studied for acquired*

carbapenem resistance. Meanwhile, these resistance traits have been poorly explored in Enterobacter species. Accordingly, two E. hormaechei strains, named Eh8322_LBHALD and Eh202_LBHALD, were identified as XDR and carbapenemase producers and, therefore, were selected for WGS.

7. Line 170-180 The ANI is often used to measure whether or not it is the same genus. SNPs are recommended to measure relatedness between strains. The authors should further calculate the number of SNPs in the core genome between ST182 strains.

R. We have removed all the ANI values in the manuscript. In addition, we have calculated number of SNPs in the core genome between ST182 strains, and included SNP pair counts and core genome identity percentage in the manuscript. Lines 181-184; 186-197; 379-381.

8. The discussion section is too lengthy, and the author narrates everything relevant in the results section in the discussion. I was unable to grasp what the central point was in the author's discussion. The authors also did not state the shortcomings and limitations of this study in the discussion section.

R. Thank you for your valuable feedback. In response to your comments, we have thoroughly revised the entire Discussion section to improve clarity and focus. The updated version avoids repeating the detailed descriptions already presented in the Results section and instead emphasizes the key findings, their implications, and comparisons with relevant literature. Additionally, we have included a new paragraph explicitly addressing the limitations of the study, as suggested (293-304). We hope these changes enhance the overall quality and readability of the manuscript.

Minor comments :

1. Line104-109 The authors' interpretation of the drug sensitivity results was incomplete, and the 34 drugs were suggested to be described in order according to drug classification (e.g., carbapenems, cephalosporins, quinolones, etc.). It is also recommended that the drugs in Fig1 be similarly categorised.

R. We appreciate the suggestion. We have revised the manuscript to provide a more complete interpretation of the antimicrobial susceptibility results, organizing the 34 antimicrobials by

drug class. Additionally, we have updated Figure 1 to reflect this classification. These changes can be found in Lines 114–125 and in the revised Figure 1.

2. Line 121 "p" should be italicized.

R. We have italicized the “p” as recommended (Line 139).

3. Fig 2B contains the contents of Fig 2A, so Fig 2A is unnecessary. The same is true for Fig 4A and Fig 4B. Fig 2A and Fig 4A should be deleted.

R. We deleted Fig 2A and Fig 4A. Thank you for your remark.

4. Line 152-156 The two strains have similar chromosomes and do not need to be described separately, which is too wordy.

R. We have deleted the description of one of the two similar chromosomes (Lines 150-156 and 172-173).

Re: Spectrum00780-25R1 (**Clinical carbapenem-resistant *Enterobacteriales* in a university hospital in Dakar, Senegal: genomic insights into *Enterobacter hormaechei* ST182 strains carrying *bla*_{NDM-5} and *bla*_{OXA-48} genes**)

Dear Dr. Komla Mawunyo Dossouvi:

Your manuscript has been accepted, and I am forwarding it to the ASM production staff for publication. Your paper will first be checked to make sure all elements meet the technical requirements. ASM staff will contact you if anything needs to be revised before copyediting and production can begin. Otherwise, you will be notified when your proofs are ready to be viewed.

Sincerely,
Felix Toka
Editor
Microbiology Spectrum

Reviewer #2 (Comments for the Author):

I would like to thank the author for carefully revising the manuscript in response to my suggestions. Their meticulous and diligent work ethic is highly commendable. However, there are still a few minor issues with the manuscript.

1. The presentation of Figure S1 is still not easy to understand. The stacked bar chart shows percentages exceeding 100%, which is clearly unconventional. We recommend that the authors use species as the coordinate axis and calculate the prevalence of AMR genes in each species, preferably including the overall prevalence. Such data would be consistent with the results presented in conventional studies.

I would like to thank the author for carefully revising the manuscript in response to my suggestions. Their meticulous and diligent work ethic is highly commendable. However, there are still a few minor issues with the manuscript.

1.The presentation of Figure S1 is still not easy to understand. The stacked bar chart shows percentages exceeding 100%, which is clearly unconventional. We recommend that the authors use species as the coordinate axis and calculate the prevalence of AMR genes in each species, preferably including the overall prevalence. Such data would be consistent with the results presented in conventional studies.

I would like to thank the author for carefully revising the manuscript in response to my suggestions. Their meticulous and diligent work ethic is highly commendable. However, there are still a few minor issues with the manuscript.

1.The presentation of Figure S1 is still not easy to understand. The stacked bar chart shows percentages exceeding 100%, which is clearly unconventional. We recommend that the authors use species as the coordinate axis and calculate the prevalence of AMR genes in each species, preferably including the overall prevalence. Such data would be consistent with the results presented in conventional studies.